# *Legionella pneumophila* and Free-Living Nematodes: Environmental Co-Occurrence and Trophic Link

**DOI:** 10.3390/microorganisms11030738

**Published:** 2023-03-13

**Authors:** Christin Hemmerling, Aurélie Labrosse, Liliane Ruess, Michael Steinert

**Affiliations:** 1Institute of Biology, Ecology, Humboldt Universität zu Berlin, Philippstraße 13, 10115 Berlin, Germany; christin.hemmerling.1@hu-berlin.de; 2Institute of Microbiology, Technische Universität Braunschweig, Spielmannstraße 7, 38106 Braunschweig, Germany

**Keywords:** *Legionella pneumophila*, free-living nematodes, *Plectus similis*, *Plectus* sp., *Acanthamoebae castellanii*, pharyngeal pumping, cooling towers, swimming lakes, ProA

## Abstract

Free-living nematodes harbor and disseminate various soil-borne bacterial pathogens. Whether they function as vectors or environmental reservoirs for the aquatic *L. pneumophila*, the causative agent of Legionnaires’ disease, is unknown. A survey screening of biofilms of natural (swimming lakes) and technical (cooling towers) water habitats in Germany revealed that nematodes can act as potential reservoirs, vectors or grazers of *L. pneumophila* in cooling towers. Consequently, the nematode species *Plectus similis* and *L. pneumophila* were isolated from the same cooling tower biofilm and taken into a monoxenic culture. Using pharyngeal pumping assays, potential feeding relationships between *P. similis* and different *L. pneumophila* strains and mutants were examined and compared with *Plectus* sp., a species isolated from a *L. pneumophila*-positive thermal source biofilm. The assays showed that bacterial suspensions and supernatants of the *L. pneumophila* cooling tower isolate KV02 decreased pumping rate and feeding activity in nematodes. However, assays investigating the hypothesized negative impact of *Legionella*’s major secretory protein ProA on pumping rate revealed opposite effects on nematodes, which points to a species-specific response to ProA. To extend the food chain by a further trophic level, *Acanthamoebae castellanii* infected with *L. pneumphila* KV02 were offered to nematodes. The pumping rates of *P. similis* increased when fed with *L. pneumophila*-infected *A. castellanii*, while *Plectus* sp. pumping rates were similar when fed either infected or non-infected *A. castellanii.* This study revealed that cooling towers are the main water bodies where *L. pneumophila* and free-living nematodes coexist and is the first step in elucidating the trophic links between coexisting taxa from that habitat. Investigating the *Legionella*–nematode–amoebae interactions underlined the importance of amoebae as reservoirs and transmission vehicles of the pathogen for nematode predators.

## 1. Introduction

The rod-shaped Gram-negative aquatic bacterium *Legionella pneumophila* is an opportunistic human pathogen and the causative agent of Legionnaire’s disease, a potentially severe pneumonia [1]. Upon inhalation of *L. pneumophila*-contaminated aerosols, the bacteria infect and replicate primarily within alveolar macrophages [2,3]. *L. pneumophila* is ubiquitous in nature, where it occurs in both natural freshwater habitats such as lakes and rivers and man-made water systems such as air conditioning units, shower heads, whirlpools and spa baths [4,5,6]. However, the majority of *L. pneumophila*-induced outbreaks of pneumonia have their origin in technical water systems, as bacterial replication is promoted by elevated temperatures [6,7,8]. In Germany, major outbreaks (2010 and 2013) were traced back to cooling towers [9,10]. Cooling towers pose a particular health risk as they can disperse contaminated aerosols over long distances, with documented infections up to 15 km away from the source [8].

In its aquatic environment, *L. pneumophila* exists as planktonic bacteria or resides as sessile cells in biofilm communities [11]. Biofilms are defined as complex microbial communities irreversibly attached to a substratum or phase boundary or to each other, embedded in a secreted matrix of extracellular polymeric substances (EPS) [12]. The microbial composition depends on abiotic factors such as water flow, temperature and light exposure, making biofilms a microbially diverse ecosystem harboring a variety of bacteria, fungi, protozoa and algae [13]. In this environment, *L. pneumophila* infects and multiplies in a wide range of amoebae (e.g., *Acanthamoeba, Naegleria* and *Vermamoeba)* and ciliates (e.g., *Tetrahymena)* [14,15,16]. Beside protists, freshwater and biofilm environments are typically populated with high densities of free-living nematodes with more than one million individuals per m^2^ [17,18]. However, nematodes are also abundant in cooling towers, where they can constitute over 80% of the eukaryotic population [19]. In their aquatic habitat, nematodes influence key biofilm processes such as oxygen turnover and the release of secondary metabolites [20,21]. Based on specific life strategies, nematode families can be assigned to a colonizer—persister scale (*c-p* scale) ranging from 1 (extreme colonizers) to 5 (extreme persisters) [22]. Colonizers (*r*-strategists) have a short life cycle, a high tolerance to disturbances and a high colonization ability [23]. Especially enrichment opportunists (*c-p* 1) show a strong population growth under food-rich conditions, while general opportunists (*c-p* 2) have a lower fecundity but higher tolerance to disturbances than the *c-p* 1 group [24]. On the other hand, persisters (*K*-strategists) typically have a long life cycle, a low colonization ability and are susceptible to disturbance [23].

A major role for the pathogenesis and ecology of *L. pneumophila* comes from secreted secondary metabolites. The export of these effector proteins into the extracellular milieu and/or target cells is mediated by the type II secretion (T2SS) and the type IV secretion system (T4SS) [25]. T4SS secretes more than 300 effector proteins, which manipulate pivotal host processes including autophagy, death pathways, protein translation and turnover and innate immunity [3]. T2SS, which also contributes to *L. pneumophila* pathogenesis, is critical for survival in water, promoting biofilm formation, sliding motility and infection of amoebal hosts such as *Acanthamoeba castellanii* and *Vermamoeba vermiformis* [26,27,28]. One of the >25 effectors secreted by T2SS is the zinc metalloprotease ProA, the major secretory protein of *L. pneumophila* [29,30]. As a virulence factor, ProA regulates the activation of other T2SS effectors, cleaves the structural proteins gelatin and casein and degrades various cytokines; recently, the degradation of collagen IV has also been demonstrated [25,31,32,33]. Furthermore, ProA is required for the optimal infection of the amoebae *Naegleria lovaniensis* and *V. vermiformis*, but interestingly not *A. castellanii* [28,34].

Recently, *L. pneumophila* was reported to infect the model nematode *Caenorhabditis elegans* as well as nematodes from environmental samples, and, to establish replicative niches within the intestinal lumen, the gonadal tissue and pseudocoelomic cavity of its nematode host [35,36]. Nematode infection with *L. pneumophila* includes a shortened lifespan, extrusion of viscera through the vulva and intestinal and anal distensions [37]. Generally, free-living nematodes are known to ingest, harbor and disseminate bacteria, that survive the gut passage, including human pathogens such as *Escherichia coli* O157:H7, *Listeria welshimeri*, *Pseudomonas aeruginosa*, *Salmonella enterica* and *Serratia marcescens* [38,39,40]. Accordingly, the role of nematodes as vectors for human pathogens has long been recognized [41,42].

The nematode diet consists of a variety of microorganisms such as algae, fungi, bacteria and protozoa, with bacteria as the main prey for aquatic nematodes [43,44]. A prerequisite for *L. pneumophila* dissemination by nematodes is a reasonable trophic linkage, i.e., feeding activity, engulfment, or attachment of the bacterium. Moreover, the fitness of the nematodes should not be affected negatively by the consumption of the pathogenic bacteria, e.g., via effector proteins. A suitable approach to address this is the pharyngeal pumping assay, which is frequently used to investigate food uptake by *C. elegans* in relation to hormones such as eicosanoids, neuromodulators such as serotonin and octopamine, drugs and microplastics [45,46,47,48]. The nematode pharynx is a neuromuscular organ, which connects the buccal cavity with the intestine [49]. This highly muscular tube shows swollen (bulb) and narrower (called propharynx or isthmus) proportions depending on nematode taxa [50]. The pharynx activity shows two distinct muscle movements: Firstly, the pharynx pumps the food into the buccal cavity and accumulates it in its anterior part. Secondly, peristalsis transports the accumulated food further on to the grinder in the terminal bulb, which crushes the food before it is passed into the intestine [51,52].

To obtain a better understanding of the interspecific interactions of *L. pneumophila* and free-living nematodes in their natural environment, the current study investigates their co-occurrence in biofilms of technical (cooling towers) and natural (swimming lakes) water habitats. This comprehensive field survey revealed that in cooling towers, nematodes could act as potential reservoirs, vectors or grazers of *L. pneumophila*. Based on this, the nematode species *Plectus similis* (Zell, 1993) and *L. pneumophila*, isolated from biofilms of a cooling tower, were cultured. To examine the feeding relationship, pharyngeal pumping assays with *P. similis* from a cooling tower and *Plectus* sp., a species isolated from a thermal spa biofilm, where it co-occurred with *L. pneumophila* [36], were performed. Different *L. pneumophila* variants (cooling tower isolate KV02, Corby wild-type, Corby Δ*proA* mutant, complementary mutant strain Corby Δ*proA proA*) were tested against *E. coli* OP50 as a control diet. Additionally, the fitness of *P. similis* and *Plectus* sp. incubated with the respective *L. pneumophila* supernatants was analyzed. A further pumping assay investigated the nematode response to *A. castellanii* infected with *L. pneumophila* KV02.

This study revealed that a *L. pneumophila* diet impairs the feeding activity in nematodes compared to *E. coli* and *A. castellanii* diets. The impact of the *Legionella* effector protein ProA on nematode predators varied with diet and nematode species. Finally, *Legionella*-infected amoebae as nematode prey seemed to act as a “Trojan horse”, facilitating bacterial transmission into nematodes.

## 2. Materials and Methods

### 2.1. Field Sampling Campaign

Swimming lakes in the Berlin–Brandenburg region in Germany were investigated for biofilms in August 2018. In 9 out of the 26 lakes surveyed, considerable biofilms were detected. In each of these lakes, the biofilms in three subhabitats were sampled: (1) water surface and algae, (2) reed or macrophytes and (3) submerged stones or litter. Water with biofilms from the different habitats was sampled in 1 l plastic flasks. The samples were transferred to cooling boxes and shipped to Humboldt-Universität Berlin for further processing. Subsamples (volume 173 cm^3^, including biofilm) were extracted using a modified Bearman method [53]. Nematodes were extracted at room temperature for 24 h, followed by a gradual heating regime for 6 h in 5 °C steps starting at 20 °C and ending at 45 °C. Afterwards, nematodes were fixed in a 5% formaldehyde solution and stored in vials at 8 °C until identification.

A total of seven *Legionella*-positive cooling towers (CT 1-7) in the Lower Saxony region in Germany were sampled in September 2018. Nematodes associated with *L. pneumophila* in a thermal source in Aix les Bains, France, were sampled in 2016. Processing details were presented in a previous work [36]. Briefly, water was sampled in 1 l sterile bottles. The biomass was collected by filtration (0.45 µm pores) and DNA was extracted using the DNeasy Power water kit from QIAGEN (Cat. No. 14900-100-NF). To verify species identity of the environmental isolate *L. pneumophila* strain KV02, PCR amplification and subsequent sequencing of the 16S rRNA gene was performed using the standard primers 27f (5′-AGAGTTTGATCCTGGCTCAG-3′) and 1492r (5′-TACGGCTACCTTGTTACGACT-3′). Resulting sequences were analyzed via NCBI Blast^®^ using the database for 16S ribosomal RNA sequences (Bacteria and Archaea).

### 2.2. Determination of the Nematode Fauna

Total nematode numbers were counted using a light microscope at 40× magnification (Leitz DIAPLAN, Leitz, Germany). Of each sample, 100 individuals were determined to the genus level at 1000× magnification (Axio Scope.A1, Zeiss, Oberkochen, Germany). In samples containing less than 100 individuals, all specimens were identified. Pictures of nematodes were taken with a five-megapixel digital camera (AxioCam ICc 5, Zeiss, Oberkochen, Germany) coupled with a picture-imaging software (ZEN (blue edition), Zeiss, Oberkochen, Germany) to assess body measurements for morphological identification. Nematodes were identified using the identification keys by Bongers [50], Holovachov and Boström [54], Zell [55] and Andrassy [56], and trophic groups, i.e., plant feeders, fungal feeders, bacterial feeders, omnivores and predators were assigned [57].

### 2.3. Cultivation of Bacteria, Amoebae and Nematodes

#### 2.3.1. Bacteria

The *L. pneumophila* isolate KV02 was isolated from CT 3 during the sampling campaign in 2018. The isolate was transformed with pXDC-50 (mCherry) kindly provided by Dr. Xavier Charpentier [58] as described previously [36]. *L. pneumophila* KV02mCherry, *L. pneumophila* Corby wild-type, *L. pneumophila* Corby Δ*proA* and *L. pneumophila* Corby Δ*proA proA* were grown on buffered charcoal–yeast extract (BCYE) agar at 37 °C for 2 days. The BCYE agar was supplemented with 12 µg/mL chloramphenicol (KV02mCherry, Corby Δ*proA proA*) and/or 20 µg/mL kanamycin (Corby Δ*proA*, Corby Δ*proA proA*). The construction and verification of the mutant strains *L. pneumophila* Corby Δ*proA* and Corby Δ*proA proA* were described in Scheithauer et al., 2021 [32].

The *Escherichia coli* strains OP50 and BL21 were grown on lysogeny broth (LB) agar at 37 °C overnight. Additionally, *E. coli* BL21 was transformed with a purified vector plasmid to enable the recombinant production of ProA. For this, the *proA* gene from *L. pneumophila* Corby was amplified with the primers proA_fw and proA_rv. The PCR fragment was then integrated into the plasmid pET22b(+) after digestion with the restriction enzymes Ndel and Xhol (for details see Scheithauer et al. [32]). The *E. coli* BL21 pET22b(+)-*proA* strain was cultured on LB agar (supplemented with 1 nM IPTG) at 37 °C overnight. The functional activity of the *proA* gene in *E. coli* BL21 pET22b(+)-*proA* was demonstrated by Scheithauer et al., 2021 [32].

#### 2.3.2. Amoebae

The axenic strain *Acanthamoeba castellanii* (ATCC 30234) was grown in 20 mL peptone yeast extract glucose broth (PYG) in cell-culture flasks for 2 days at 37 °C and 5% CO_2_ before further procession.

#### 2.3.3. Nematodes

*Plectus similis* from CT 3 and *Plectus* sp. from the thermal source in Aix les Bains, France, were maintained on Page’s Amoeba saline (PAS) agar at 20 °C with no additional food source. For the following assays, nematodes were age-synchronized by screening the population on the PAS plates using a binocular and hand-transferring only adult individuals to the assay plates.

### 2.4. Confocal Laser-Scanner Microscopy

*P. similis* and *Plectus* sp. were inspected for the ingestion of *L. pneumophila* via confocal laser-scanning microscopy (CLSM; Leica TCS SP8, Wetzlar, Germany). For this, *L. pneumophila* KV02mCherry were adjusted to 3 × 10^4^ colony-forming units /mL (cfu/mL) in PAS buffer and transferred to an 8-well ibidi^®^ plate. Then, nematodes were added to the *Legionella* suspension. After incubation for 72 h in the dark at room temperature, samples were fixed with 20 µL of 16% paraformaldehyde and nematodes were microscopely scanned for the ingestion of bacterial cells.

### 2.5. Infection of A. castellanii

After amoebae reached confluency after incubation for 2–3 days at 37 °C, adherent cells were centrifuged at 233× *g* for 20 min and adjusted to 1 × 10^5^ cfu/mL with PAS buffer. 5 mL of the suspension were transferred in cell-culture flasks and incubated for 30–60 min at 37 °C. Then, adherent cells were infected with *L. pneumophila* KV02mCherry. Therefore, freshly grown colonies of *L. pneumophila* KV02mCherry were resuspended in phosphate-buffered saline (PBS) and adjusted to 1 × 10^6^ cfu/mL. Then, amoebae were infected with 50 µL of the adjusted bacteria suspension at an MOI (multiplicity of infection) of 0.02 and incubated for 24 h at 37 °C before further procession.

### 2.6. Pharyngeal Pumping Assay

#### 2.6.1. Experimental Set-Up

To test the impact of bacteria (*L. pneumophila* and *E. coli* strains) and amoebae (*A. castellanii*) as food source on the pharyngeal pumping frequency of the free-living nematodes *P. similis* and *Plectus* sp., assay plates were prepared as followed: 300 µL of either bacteria or amoebae suspension was seeded onto nematode growth medium (NGM) agar plates. After the suspension was dried, 140 adult nematodes were hand-picked from a PAS agar plate onto an assay plate (3 replicates with *n* = 140 individuals). The nematodes were then allowed to adapt before being tested.

The optimal bacteria and amoebae concentrations and nematode adaption time were determined in preliminary tests. Therefore, suspensions with different concentrations (for bacteria 1 × 10^4^, 10^6^ and 10^8^ cfu/mL and amoebae 1 × 10^3^, 10^4^ and 10^5^ cfu/mL) were seeded onto NGM plates (3 replicates per concentration). Then, 20 nematode individuals (originating from a PAS cultivation plate) were placed onto each assay plate and the nematode activity and feeding behavior was monitored every hour over a period of eight hours. The concentration and time at which the highest proportion of individuals was feeding was chosen for the pumping assay.

As the pumping rate of nematodes is too fast to count correctly in real time, 1 min videos of individuals feeding (500× magnification) were recorded using a VHX-600 digital microscope (Keyence Corporation, Osaka, Japan). For analysis, videos were then played back at half speed to count each individual pump. Generally, videos of 15 animals were recorded per replicate. However, as the tested strains had different effects on the feeding behavior, it was not always possible to record as much as 15 feeding individuals for each treatment and replicate. The *L. pneumophila* diets resulted in an especially high inactivity of nematodes (presumably caused by toxic secondary metabolites released by the offered bacteria), resulting in less than 15 feeding individuals observed per replicate. Thus, pumping rates were reported as contractions min^−1^ per individual.

#### 2.6.2. Food Resources

Freshly grown colonies of *L. pneumophila* strains, i.e., *L. pneumophila* isolate KV02mCherry, *L. pneumophila* Corby wild-type and *L. pneumophila* Corby Δ*proA*, were resuspended in PBS buffer. A final concentration of 1 × 10^6^ cfu/mL was seeded onto the assay plates and nematode adaption time was 3 h.

Fresh *E. coli* OP50 colonies were resuspended in PBS buffer and offered to nematodes in a concentration of 1 × 10^6^ cfu/mL. Nematode adaption time was 8 h.

24 h after infection with *L. pneumophila* KV02mCherry, *A. castellanii* were seeded onto the assay plates. As a control, nematodes were fed with non-infected *A. castellanii*. Adaption time was 8 h for both treatments.

#### 2.6.3. Effector Proteins

To test whether the negatively affected nematode-feeding behavior is due to toxic effector proteins released from *L. pneumophila,* rather than the bacterial cells themselves, *E. coli* OP50 cells were added to supernatant of the *L. pneumophila* isolate KV02mCherry. For this, fresh *L. pneumophila* KV02mCherry colonies were resuspended in PBS buffer (1 × 10^6^ cfu/mL) and incubated for 24 h at 4 °C to allow the release of bacterial compounds into the buffer. Then, the *L. pneumophila* KV02mCherry suspension was centrifuged at 4000× *g* for 20 min. The pellet was discarded and the supernatant was sterile-filtered through one layer of 0.2 µm filter paper. The resultant filtrate was tested for the absence of any cells by plating onto BCYE agar. In order to add the filtrate to *E. coli* OP50, *E. coli* OP50 in LB medium (1 × 10^6^ cfu/mL) was centrifuged at 3000× *g* for 20 min. The supernatant was discarded and the *E. coli* OP50 pellet was resuspended in *L. pneumophila* filtrate. The final suspension was seeded onto the assay plates and nematode adaption time was 3 h.

In a next step, the collagen IV-degrading protease ProA was tested for its negative effect on feeding activity. Therefore, ProA was isolated from *L. pneumophila* Corby supernatant and purified after a previously described method [32]. The functional activity of purified ProA in degrading collagen IV was demonstrated by Scheithauer et al., 2021 [32]. Then, nematodes were incubated in 10 µg/mL purified ProA in cryo-DEAE buffer for 2 h. After incubation, nematodes were transferred onto the assay plate prepared with a dried *E. coli* OP50 lawn (1 × 10^6^ cfu/mL). Adaption time was 8 h for both treatments.

Additionally, freshly grown colonies of *E. coli* BL21 pET22b(+)-*proA* were resuspended in PBS buffer and offered to nematodes in a concentration of 1 × 10^6^ cfu/mL. As a control, nematodes were fed with the native *E. coli* BL21 strain. The PBS buffer as well as the NGM agar of both treatments contained 1 mM IPTG. Adaption time was 8 h for both treatments.

### 2.7. Nematode Fitness

To test the effect of effector proteins in *Legionella* supernatant on nematode fitness, nematodes were added to supernatants of the following *L. pneumophila* strains: *L. pneumophila* KV02mCherry, *L. pneumophila* Corby wild-type, *L. pneumophila* Corby Δ*proA* and *L. pneumophila* Corby Δ*proA proA*, as controls were pure PBS buffer without *L. pneumophila* contact.

Adult nematodes (35–45 individuals per sample) were hand-picked from PAS agar plates into 15 mL Falcon tubes with 2 mL sterile tab water. After 2 h, the supernatant (1 mL) of the tap water was discarded and 1 mL fresh tap water was added as washing procedure. Then, the nematode samples were incubated overnight at room temperature. The following day, the supernatant (1 mL) of the tapb water was discarded and 2 mL of *L. pneumophila* filtrate or PBS buffer (control) was added. Each treatment was replicated four times and incubated at room temperature. After 24 h and 48 h, the number of dead nematodes was counted using a light microscope (10× magnification). Nematodes were considered to be dead if they did not move when touched with a fine needle.

### 2.8. Statistics

The data were tested for normal distribution (Shapiro–Wilk normality test) and homogeneity of variance (Levene’s test). If the test requirements were met by the data, a two-sided analysis of variance (ANOVA) followed by Tukey’s post hoc test (significance level at *p* < 0.05) was performed. If the data did not meet normal distribution, either the Mann–Whitney U test or the Kruskal–Wallis rank sum test with Dunn’s post hoc test (with Bonferroni correction and a significance level at *p* < 0.05) was performed according to the number of groups to be compared. All statistical analyses were performed with the software R (version 4.1.2 “Bird Hippie”).

## 3. Results

### 3.1. Co-Occurrence of Free-Living Nematodes and Legionella pneumophila in Natural and Technical Water Habitats

In total, only seven nematode genera were found across all investigated cooling towers (CTs, Table 1). The number of genera per individual CT ranged from two (CT 1 & 4), two (CT 3 & 5) and one (CT 6 & 7) to zero (CT 2). *Plectus* was the most abundant genus with an occurrence in four different CTs. The highest densities were reached by *Plectus* with 1045 Ind. 100^−1^ mL in CT 1 and *Diploscapter* with 162 Ind. 100^−1^ mL in CT 5. Bacterial feeders were the only trophic group present in five CTs, while in CT 1 and CT 4, the fungal feeder *Filenchus* also occurred (Table 1). Considering nematode life strategy, i.e., colonizers (*c*) or persisters (*p*) [23], the biofilms in CTs were mainly inhabited by opportunistic species with a *c-p* value of 2, i.e., general opportunists according to the *c-p* scale introduced by Bongers [23], except CT 5, where only strong *r*-strategists (*c-p* 1) were detected. *L. pneumophila* was detected in all CTs except for CT 6, where biocide shock dosage before sampling prevented its detection.

Compared to CTs, the nematode community in swimming lakes was much more diverse. The most genera were detected on biofilms around algae (14), followed by submerged stones (12) and macrophytes (11) (Table 1). The predominating genus across sub-habitats was *Chromadorina*, with densities ranging from 63 Ind. 100^−1^ mL to 136 Ind. 100^−1^ mL for algae and macrophytes, respectively. All subhabitats were characterized by a complex trophic structure with microbial feeders, omnivores and predators. Moreover, nematodes in the biofilms of swimming lakes showed a wide range of life strategies from *r*-strategists over opportunists to *K*-strategists. Three nematode genera, i.e., *Filenchus*, *Heterocephalobus* and *Plectus*, occurred in both natural and technical water habitats. *L. pneumophila* could not be detected in swimming lakes.

### 3.2. Ingestion of Legionella pneumophila by Plectus similis and Plectus sp.

Two free-living nematodes, *P. similis* as the most abundant genus across CTs and *Plectus* sp. from a thermal spa biofilm, were chosen as model candidates for further experiments. In a first step, *P. similis* and *Plectus* sp. were tested for their general ability to take up *L. pneumophila.* For this, nematodes were fed with the cooling tower isolate *L. pneumophila* KV02. Species identification of the isolate KV02 was confirmed by 16S rRNA gene sequencing with an alignment of 99.86%. Following the transformation of KV02 with mCherry (red fluorescence), confocal laser-scanner microscopy revealed the presence of KV02mCherry in different regions within the pharynx and intestine of nematodes (Figure 1). In *P. similis,* a single rod-shaped KV02mCherry cell was located right before the grinder within the terminal bulb (Figure 1A). In *Plectus* sp., KV02mCherry cells could be detected within the propharynx, grinder and upper intestine (Figure 1B), as well as mid-gut (Figure 1C).

### 3.3. Legionella pneumophila Affects Pharyngeal Pumping Activity in Plectus similis and Plectus sp.

After microscopic confirmation of *L. pneumophila* ingestion by *P. similis* and *Plectus* sp., the effect of *L. pneumophila* KV02mCherry on the pharyngeal pumping activity of nematodes in comparison to *E. coli* OP50 was determined (Figure 2). *P. similis* fed with OP50 showed, with 135 contractions min^−1^, a 70% higher pumping frequency compared to KV02mCherry, with 80 contractions min^−1^ (*p* < 0.001) (Appendix A). Similarly, *Plectus* sp. displayed, with 114 contractions min^−1^, a three-times-higher pumping frequency when fed with OP50 (*p* < 0.001) compared to *Legionella* (Appendix A).

As the pumping rates differed clearly between bacteria species, we hypothesized that the effector proteins secreted by KV02mCherry manipulate the hosts’ feeding activity, inducing an impaired pharyngeal activity. To test this hypothesis, nematodes were fed with OP50 mixed with KV02mCherry supernatant (Figure 2). In this set-up, the pumping frequency of *P. similis*, with 44 contractions min^−1^, was 16% higher than when fed with KV02mCherry directly, but still significantly lower (*p* < 0.001) compared to the sole OP50 diet. A similar picture was displayed by *Plectus* sp. The pumping frequency, with 44 contractions min^−1^ when exposed to OP50 plus supernatant, increased by 33% compared to KV02mCherry cells, but this was again significantly slower (*p* < 0.001) than feeding OP50.

### 3.4. Effector Protein ProA as a Prospect Candidate for the Regulation of Pumping Activity

The experiments with the addition of *L. pneumophila* KV02mCherry supernatant to *E. coli* OP50 point to an impaired pumping frequency of nematodes (Figure 2). One of the most abundant effector proteins in *Legionella* supernatant is the zinc metalloprotease ProA, with its broad proteolytic and cytotoxic activities [29,33]. ProA degrades collagen IV, which is a major component of the basement membranes covering the pharynx, intestine and gonads of nematodes [32,59]. Consequently, ProA was investigated for its potential to alter pumping activity.

In a first step, pumping frequencies were compared between *L. pneumophila* Corby wild-type and *L. pneumophila* Corby Δ*proA*, a mutant, which does not produce this protease (Figure 3). When fed with the Corby Δ*proA* mutant, *P. similis* had, with 100 contractions min^−1^, a higher pumping frequency compared to feeding Corby wild-type with 61 contractions min^−1^. Contrary to *P. similis*, the pumping activity of *Plectus* sp. was low when offering the Corby Δ*proA* mutant (25 contractions min^−1^), but increased to 43 contractions min^−1^ with Corby wild-type. Additionally, to guarantee that the complementation of the Corby Δ*proA* deletion rescued the observed phenotype, the complement mutant strain *L. pneumophila* Corby Δ*proA proA* was tested (Figure 3). The pumping frequency of *P. similis* in the presence of the complement mutant strain *L. pneumophila* Corby Δ*proA proA* (75 contractions min^−1^) was in between the frequency in presence of the wild strain (61 contractions min^−1^) and the frequency with the Corby Δ*proA* mutant (100 contractions min^−1^). This tendency was not observed for *Plectus* sp., as the frequency in the presence of both mutant strains was identically reduced to 25 contractions min^−1^.

To further analyze the effect of ProA, nematodes were incubated in a cryo-DEAE buffer with 10 µg/mL purified ProA isolated from *L. pneumophila* Corby wild-type for 2 h, before offering *E. coli* OP50 as a diet (Figure 3). Only *Plectus* sp. responded to this treatment with pumps significantly lower (*p* < 0.05) in the group incubated with ProA, with 90 contractions min^−1^, than in the group fed with sole *E. coli* OP50, pumping with 114 contractions min^−1^. Additionally, a negative control of nematodes incubated in sole cryo-DEAE buffer was performed to exclude the potential negative effects of the cryo-DEAE buffer on the pharyngeal pumping activity.

In a next step, nematodes were fed either with the *E. coli* strain BL21 pET22b(+)-*proA* or the *E. coli* strain BL21 not capable of ProA synthesis. This set-up examines whether pharyngeal pumping of nematodes is affected by the protease, if synthetized by *E. coli* as bacterium with otherwise good food quality. For *P. similis*, pumping frequency was significantly reduced (*p* < 0.05), with 155 and 177 contractions min^−1^ when feeding BL21 pET22b(+)-*proA* and the control strain, respectively (Figure 3). For *Plectus* sp., the pumping activity for both strains was equal, differing by only 10 contractions min^−1^.

### 3.5. Acanthamoeba castellanii as Potential Promotor for Legionella pneumophila Ingestion

Links between *L. pneumophila* and nematodes may also include a further trophic level, i.e., free-living amoebae, which are well known as hosts for pathogenic bacteria. To investigate this “Trojan-horse” approach, nematodes were fed with *A. castellanii* 24 h after its infection with *L. pneumophila* KV02mCherry (Figure 4). The pumping frequency of *P. similis* was 34% higher (*p* < 0.001) when fed with *L. pneumophila* KV02mCherry infected amoebae (141 contractions min^−1^) compared to non-infected ones (105 contractions min^−1^). In contrast, for *Plectus* sp., no difference in pharyngeal pumping between amoebae with/without *L. pneumophila* KV02mCherry was detected.

### 3.6. Influence of Diet on Overall Nematode Feeding Behavior

To determine the impact of different diets on nematode behavior, all individuals on each assay plate were counted after a respective adaption time, i.e., after 3 h for *Legionella* or 8 h for *E. coli* and *A. castellanii.* The following three behaviors were distinguished: (1) “inactive” nematodes with no apparent movement, (2) “active” fast-moving nematodes freely roaming the bacterial lawn and (3) “feeding” nematodes dwelling in a small area on the bacterial lawn, visibly pumping with their pharynx.

For *P. similis*, when fed with *L. pneumophila*, nematodes were most active (83%) with Corby Δ*proA proA* as diet and least active (41%) with Corby wild-type as diet (*p* < 0.05, Table 2). Otherwise, the proportions of inactive, active and feeding individuals were similar between the different *L. pneumophila* strains. For *Plectus* sp., a comparable pattern was observed, but with a higher occurrence of inactive individuals (59–77%).

With *E. coli* OP50 as diet, the proportions of active and feeding *P. similis* where highest (87 and 88%, respectively) when fed with OP50 and lowest (24 and 12%, respectively) when individuals were incubated with ProA prior to the assay (*p* < 0.05, Table 2). On the other hand, significantly more nematodes were inactive (76%) due to preincubation with ProA than without (12%, *p* < 0.05). For *Plectus* sp., the activity mode of nematodes was similar between all *E. coli* OP50 treatments. However, the proportion of feeding individuals was significantly higher (*p* < 0.05) when fed with OP50 (39%) than when fed with OP50 mixed with KV02mCherry filtrate (11%).

The synthesis of the protease ProA by *E. coli* did not affect nematode behavior. The proportions of inactive, active and feeding individuals of both *P. similis* and *Plectus* sp. were nearly equal between *E. coli* BL21 pET22b(+)-*proA* and the control diet *E. coli* BL21 (Table 2).

For *P. similis*, when fed with *A. castellanii*, the proportions of inactive, active and feeding individuals, ranging between 41% to 56%, were similar for infected and non-infected amoebae (Table 2). Comparably, in *Plectus* sp., no significant response to amoebae carrying *L. pneumophila* or not was observed. However, the overall behavior to the amoebae as prey differed to that of *P. similis*. The proportion of active individuals of *Plectus* sp. was highest (62–69%), and that of inactive nematodes was lowest (31–38%). This difference was significant (*p* < 0.05) for infected amoebae as diet.

To assign differences in the general feeding behavior of both nematodes related to the four different diet groups, the wild-type strains *E. coli* OP50, *E. coli* BL21 and *A. castellanii* were statistically compared to *L. pneumophila* KV02mCherry. The pumping activity of *P. similis* was significantly lower when fed with *L. pneumophila* KV02mCherry compared to *E. coli* OP50 (*p* < 0.05). For *Plectus* sp., the activity was significantly lower when fed with *L. pneumophila* KV02mCherry compared to *A. castellanii* (*p* < 0.05).

### 3.7. Effect of Legionella pneumophila Supernatant on Nematode Fitness

As *E. coli* OP50 mixed with *L. pneumophila* KV02mCherry filtrate as diet resulted in impaired pharyngeal pumping frequency, the effects of *Legionella* supernatant on nematode fitness were studied more thoroughly. Nematodes were incubated in four different supernatants or PBS solely as control, and the mortality was determined after 24 h and 48 h. This revealed that in general, *P. similis* had a higher overall mortality when incubated with *Legionella* supernatant than *Plectus* sp. (Figure 5).

The mortality rate of *P. similis*, ranging from 9% (PBS) to 24% (Corby Δ*proA*), was more than twice as high for *Legionella* (except for Corby Δ*proA proA*) supernatant than for the PBS control after incubation for 24 h (Figure 5). This was significant for the supernatants of Corby Δ*proA* (*p* < 0.01) and Corby wild-type (*p* < 0.05). Additionally, the mortality rate of *P. similis* was significantly lower for the complementary mutant Corby Δ*proA proA* when compared to Corby Δ*proA* (*p* < 0.01) and Corby wild-type (*p* < 0.05). After 48 h, the mortality of Corby Δ*proA* and Corby wild-type supernatant increased by 25% and 45%, respectively, compared to 24 h. Both supernatants increased mortality significantly (*p* < 0.05) compared to PBS after 48 h. On the other hand, the mortality of Corby Δ*proA proA* decreased by half compared to 24 h.

*Plectus* sp. mortality ranged between 8% (Corby Δ*proA proA*) and 20% (Corby Δ*proA*) after incubation with the respective supernatant for 24 h (Figure 5). Compared to PBS, the mortality of *Plectus* sp. increased by about one half to 14% and 15% in the supernatants of KV02mCherry and Corby wild-type, respectively. Further, the mortality of *Plectus* sp. increased to 20% in the supernatant of the Corby Δ*proA* mutant. Moreover, the mortality of the Corby Δ*proA* supernatant was significantly higher than the mortality of Corby Δ*proA proA* supernatant (*p* < 0.05). After 48 h, the picture was different: the mortality of nematodes was high in all treatment variants, including the control. As at 24 h, most dead individuals were detected after incubation with Corby Δ*proA* (27%); however, the lowest numbers occurred with Corby Δ*proA proA* supernatant (13%, *p* < 0.01), Corby wild-type (15%, *p* < 0.05) and KV02mCherry (17%, *p* < 0.05) supernatant. Additionally, with PBS, more than twice as many individuals died at 48 h compared to 24 h.

## 4. Discussion

### 4.1. Free-Living Nematodes and L. pneumophila Co-Occur in Cooling Towers

This survey, investigating natural (swimming lakes) and technical (cooling towers) water bodies, revealed the co-occurrence of *L. pneumophila* and free-living nematodes within cooling towers. Recently, this was also reported by another cooling-tower study [19]. However, *L. pneumophila* and nematodes were not associated together in natural swimming lakes, but co-occurred in a natural thermal source.

Cooling towers provide a unique environment for microbial growth, considering temperatures ranging between 25 °C and 35 °C, a neutral pH, and continuous aeration [60]. On the other hand, the high operating temperatures, in combination with the application of biocides, play an important selective role on all biota inhabiting the water body of cooling towers [61]. This was distinctly visible in the extremely low diversity of the nematode community, comprising one to three taxa only, all generalists that are tolerant to pollutants and other disturbances and are, moreover, able to survive food-poor conditions [62].

The widest distribution across cooling towers, with occurrences in four out of seven, showed the genus *Plectus* and taxa in the family Cephalobidae (i.e., *Acrobeloides*, *Eucephalobus*, *Heterocephalobus*). Their presence matches their opportunistic life strategy, i.e., they are adapted to exploit newly available habitats, and, with their rapid growth rate, quickly establish populations [23]. They can cope with unpredictable or variable environments such as a cooling tower, and here, most of them are additionally favored due to their capacity to tolerate temperatures well above 30 °C [63]. Cephalobidae are the most common group in soils worldwide [64], while Plectidae are also abundant in humid habitats, e.g., mosses, with proportions up to 60% of the nematode community [65,66]. This likely explains the frequent occurrence of *Plectus* in cooling-tower biofilms. The thermophile *Diploscapter* sp., with an optimal population growth at 30 °C [67], was only found in a single cooling tower, but findings from wastewater biofilms, trickling filters and activated sludge are documented [67,68]. In sum, all nematode taxa detected in cooling towers (except *Filenchus*) were bacterial feeders. Apparently, the biofilm community offers few resources, only supporting short food chains lacking large-sized omnivores and predators at higher trophic levels.

Unlike in cooling towers, biofilms in lakes are hotspots for biodiversity, with a wide variety of microorganisms and their grazers in numerous ecological niches [69,70]. However, no *L. pneumophila* could be detected in the investigated swimming lakes, although the water temperature (mean: 25.1 °C, Appendix A) during the sampling period was suitable for its growth [71,72]. In multispecies biofilms, bacteria compete for a better accessibility and utilization of nutrients, thus faster-growing species, e.g., *P. aeruginosa*, likely outcompete *L. pneumophila* [73]. Moreover, some bacteria exhibit inhibitory effects on *L. pneumophila,* e.g., *Aeromonas hydrophila*, *Acidovorax* sp. and *Sphingomonas* sp. [74]. Evidently, *L. pneumophila* can better use the framework of its ecological potency in low-diversity habitats such as cooling towers.

As for other biofilm biota, the nematode populations were distinctly more diverse in the biofilms of swimming lakes than those of cooling towers, reflecting the greater microbial diversity and the related number of trophic niches [75]. The most abundant across lake subhabitats was *Chromadorina* sp., a genus which predominates in freshwater biofilms [17,76,77]. Interestingly, taxa which occurred in both natural and technical water habitats, i.e., *Filenchus*, *Heterocephalobus* and *Plectus*, where much more abundant in the cooling towers than in swimming lakes. The cooling tower biofilms likely offered an environment with less predation and/or resource competition, fostering population growth in adapted taxa. The fact that only single or very few species were encountered an entire cooling tower habitat suggests the random, windborne dispersal of nematodes.

As in cooling towers, bacterial feeders were the dominant trophic group in all three subhabitats of the swimming lakes. However, in the lake biofilms, high numbers of omnivores and predators were also detected. These *K*-strategists, with long lifespans and high sensitivity to disturbance, reflect the balanced environmental conditions as well as stability due to a better nutrient supply in lakes [18].

### 4.2. L. pneumophila Diet Reduces Feeding Activity in Free-Living Nematodes

Free-living nematodes take up food by rhythmic contractions of the pharynx muscles, which makes these pharyngeal pumps a reliable indicator of food ingestion [78]. In turn, the availability, quality and familiarity of the food affects the rate of pharyngeal pumping [79]. Analyzing the pumping rates of *P. similis* and *Plectus* sp. revealed that both species pumped significantly more when fed with *E. coli* OP50 compared to *L. pneumophila* KV02mCherry. As pharyngeal pumping shows a graded response to food availability [80], this indicates *E. coli* OP50 as a quality resource. In contrast, slow pumping punctured with long pauses was observed for *Legionella,* a behavior typical for worms without food [80]. Comparing synthetic beads and *E. coli* OP50, Fueser et al. [47] showed that *C. elegans* restricts pumping for particles with low nutritional value to a basic rate. This behavior prevents the nematode from wasting energy by high-frequency pumping, but still allows screening for food. One factor contributing to food quality is the bacterial cell size, yet with a length of 1 to 3 µm, *Legionella* is in the same range as *E. coli* OP50 [81,82]. In conclusion, the downregulated pumping rate in *P. similis* and *Plectus* sp. suggests that *L. pneumophila* is a poor or unsuitable (e.g., well-defended—see Section 4.3) food source for *Plectus*.

The food quality can be further assessed by the foraging behavior of nematodes, which exhibits two discrete foraging states called roaming and dwelling [83]. Roaming is a rapid, straight movement to explore the environment, whereas dwelling is characterized by slow movements with frequent reversals and turns [84]. On poor food, roaming strongly increases, while dwelling predominates on high quality food [84,85]. When fed with the *E. coli* strains OP50 and BL21 or the amoeba *A. castellanii*, the nematodes moved slowly, equaling dwelling, and proportions of moving and feeding individuals were more-or-less similar (personal observation). In contrast, in the presence of *L. pneumophila* strains, foraging activity switched to an exploratory behavior, i.e., the proportion of moving individuals strongly increased compared to slow-moving, feeding specimens. This change from dwelling to roaming again points to a lower food quality of *L. penumophila* for *Plectus* compared to the common diet *E. coli,* but also to *A. castellanii*.

### 4.3. L. pneumophila Reduces Fitness of Free-Living Nematodes

Secondary metabolites secreted by bacteria can act not only as repellents but also as toxins for nematodes [86,87]. For example, the production of cyanide by *P. aeruginosa* ceases pharyngeal pumping followed by progressive paralysis and death in *C. elegans* [88]. To test whether *L. pneumophila* effectors regulate the pharyngeal pumping rate, we offered *E. coli* OP50 suspended in *Legionella* KV02mCherry supernatant as diet. Interestingly, the pumping rate of *P. similis* and *Plectus* sp. decreased significantly compared to pure *E. coli* OP50 and was almost as low as for the KV02mCherry diet. This strongly suggests that secondary metabolites in the supernatant of *L. pneumophila*, e.g., the *Legionella* major secretory protein ProA, can critically impair the pharyngeal pumping activity of *Plectus.*

Supporting this, in *P. similis*, the pumping rate was higher (albeit not significantly) when fed with a *L. pneumophila* Corby Δ*proA* mutant strain compared to the Corby wild-type and Corby Δ*proA proA* strains, respectively (Figure 3). Moreover, feeding on *E. coli* BL21 pET22b(+)-*proA* decreased the pumping rate. Due to the high standard deviations for E. coli BL21 pET22b(+)-proA and its control BL21, the statistically assigned significant reduction in the pumping rates should be regarded more as a trend. In contrast, *Plectus* sp. decreased the pumping rate when fed with the *L. pneumophila* strains Δ*proA* and Δ*proA proA*, but not with *E. coli* BL21 pET22b(+)-*proA*. However, the incubation with purified ProA also impaired the pumping rate. As mentioned above, taking into account the biological variability in pumping activity for *E. coli* OP50 + ProA and its control without ProA, this points to a trend rather than a distinct decrease in the pumping rate in the presence of ProA.

In sum, the impact of ProA on nematode feeding activity varied with diet, and further showed species-specific differences. Moreover, no direct toxicity of ProA was observed, as the incubation of nematodes with the supernatant of the *L. pneumophila* Δ*proA* strain for 24 and 48 h, respectively, did not result in a reduced mortality compared to the Corby wild-type, Corby Δ*proA proA* and KV02mCherry strains (Figure 5).

Besides impaired feeding, bacterial defense can expose negative impact on nematode fitness [89,90]. Secondary metabolites with nematicidal potential are reported for common soil and freshwater bacteria such as *Pseudomonas* e.g., [91,92]. In line with this, the mortality of *Plectus* increased after incubation in *L. pneumophila* supernatant (except for Corby Δ*proA proA*) for 24 h when compared to PBS buffer. However, the high mortality in PBS (control) after 48 h for *Plectus* sp. does not allow for a clear statement about the mortality of *Legionella* supernatant after 48 h. Besides ProA, other potentially destructive enzymes in *L. pneumophila* supernatant that degrade host cell components are T2SS-dependent proteases, peptidases, acid phosphatases, lipases, phospholipases A and C, lysophospholipase A, a cholesterol acyltransferase and an RNase [34]. Such secretions of *L. pneumophila* have the potential to reduce nematode fitness, e.g., protease and lipase activity can lead to structural damage in the cuticle, followed by a decrease in motility. Similarly, an extracellular serine protease from *Bacillus* sp. was reported to degrade the cuticle of the bacterial feeder *Panagrellus redivivus* causing the nematodes’ death [93]. Further, Siddiqui et al. [94] showed that the exposure of the root-knot nematode *Meloidogyne incognita* to culture filtrate of *Pseudomonas fluorescens* CHA0 killed nematodes, which was induced by the extracellular protease AprA. Further studies are needed to clarify which of the potential destructive secondary metabolites are the causative agent for this negative *Legionella*–nematode interaction.

### 4.4. A. castellanii as Trojan Horse for L. pneumophila Transmission

*L. pneumophila* resists degradation by amoebae and other protozoa and multiplies intracellularly [16]. Such hosts of bacterial pathogens, such as the free-living amoeba *A. castellanii,* are considered the “Trojan horse” of the microbial world [95]. Virtually nothing is known regarding whether these *Legionella*—amoeba interactions involve higher trophic levels, i.e., predators of *A. castellanii* such as nematodes.

Pharyngeal pumping assays investigating this trophic link revealed that when fed with *L. pneumophila*-infected *A. castellanii* (i.e., 24 h post infection with *L. pneumophila* KV02mCherry), the pumping rate of nematodes was similar (*Plectus* sp.) or even significantly higher (*P. similis)* compared to non-infected amoebae. The video recordings showed that the nematodes do not swallow the amoeba cells whole, but rather rupture the cell membrane and suck in the cell contents (Appendix A). It is likely that the disruption of the physical and structural integrity of the amoebal plasma membrane, facilitated by the pore-forming activity of intracellular *L. pneumophila*, allowed for easy access to amoebal cytoplasma, and in the case of *P. similis*, also stimulated pumping activity.

By their sucking feeding mode, nematodes incorporated the liberated pathogen from the amoeba diet. Similarly, Rasch et al. [36] observed *Legionella* cells, but no intact *Legionella*-infected *A. castellanii* within the digestive tract of *C. elegans,* and thus argued against a “typical” Trojan horse transmission of *L. pneumophila* via *A. castellanii* into nematodes. On the other hand, the tested *Plectus* species did not discriminate against infected amoebae, as indicated by the observed pumping rates. It is possible that *L. pneumophila* “hides” from nematode detection inside the amoeba and, in this way, could access nematodes as a new host without being detected before ingestion. Moreover, nematodes preferred *L. pneumophila*-infected amoebae over a sole *L. pneumophila* diet, pointing towards a “Trojan horse-like transmission” strategy of *Legionella* in *Plectus*.

## 5. Conclusions

This study identified cooling towers as technical water habitats, where free-living nematodes and *L. pneumophila* co-occur in biofilm communities. The species *P. similis* (coolingtower isolate) and *Plectus* sp. (thermalspa isolate) were chosen in order to examine the feeding relationship between *L. pneumophila* and nematodes. Pharyngeal pumping assays revealed that *L. pneumophila* impairs the pumping rate of *Plectus* compared to *E. coli* and *A. castellanii* diets. However, the impact of the *Legionella* major secretory protein ProA on nematode feeding activity varied with diet and nematode species.

Overall, reduced feeding activity induced by *L. pneumophila* suggests that *Legionella* is no common resource of the genus *Plectus*, as shown in an artificial set-up on agar plates. However, the ingestion of *L. pneumophila,* resulting in an intracellular infection of the nematodes’ digestive tract, can occur and has to be considered in pathogen dissemination. Moreover, trophic interactions between *Plectus*- and *Legionella*-infected amoebae point to a “Trojan horse-like transmission” of *L. pneumophila* into nematodes. In order to further decipher the *Legionella*–nematode interaction, other coolingtower taxa, e.g., the thermophilic *Diploscapter*, have to be tested for their potential as reservoirs or vectors for *L. pneumophila* in future studies. Additionally, feeding assays in (artificial) biofilms are necessary to obtain a more realistic picture on the predator–prey relationship between nematodes and *L. pneumophila*.

## Figures and Tables

**Figure 1 microorganisms-11-00738-f001:**
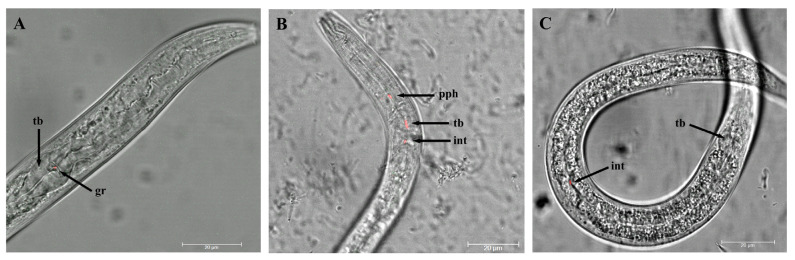
Confocal laser-scanner microscopy of *L. pneumophila* KV02mCherry (cooling tower isolate) ingested by the nematodes *Plectus similis* and *Plectus* sp. Single rod-shaped cells of *L. pneumophila* KV02mCherry are visible in red. (**A**) *P. similis* with *L. pneumophila* KV02mCherry right before the grinder (gr) located within the terminal bulb (tb). *Plectus* sp. with several bacterial cells in the propharynx (pph), terminal bulb and upper intestine (int) (**B**), as well as a single cell in the mid-gut (**C**).

**Figure 2 microorganisms-11-00738-f002:**
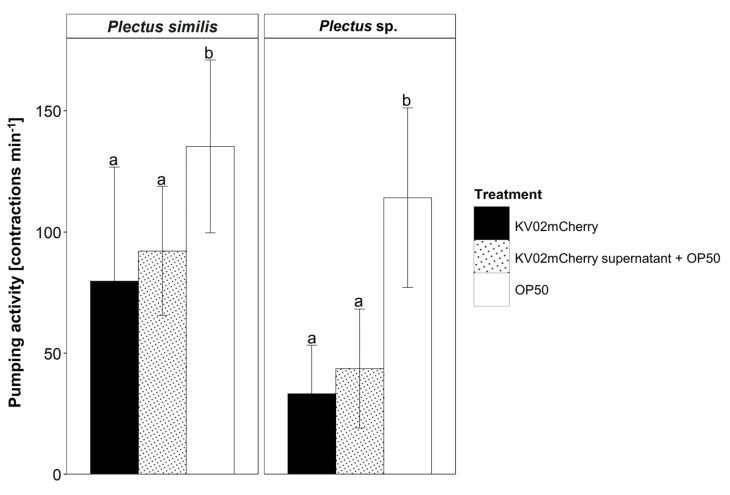
Pharyngeal pumping activity per individual [contractions min^−1^ ± SD] of the nematode species *Plectus similis* and *Plectus* sp. fed with *L. pneumophila* KV02mCherry (cooling tower isolate), *E. coli* OP50 or *E. coli* OP50 mixed with *L. pneumophila* KV02mCherry supernatant. Bars with the same letter are not statistically different according to Dunn’s post hoc test (*p* < 0.05).

**Figure 3 microorganisms-11-00738-f003:**
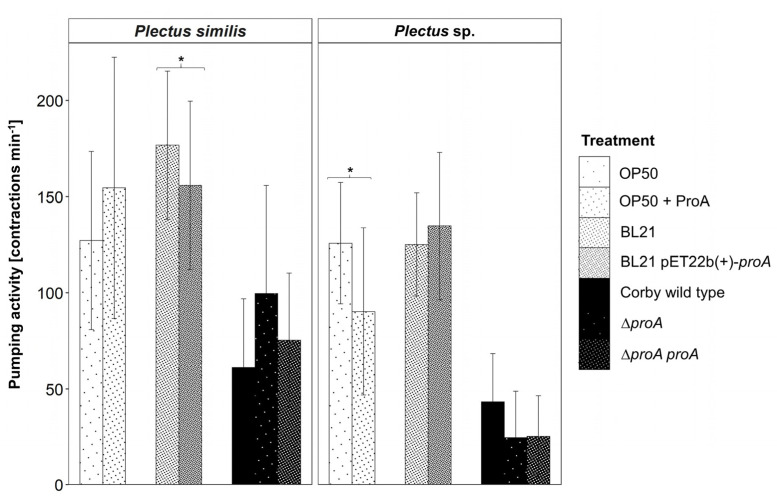
Pharyngeal pumping activity per individual [contractions min^−1^ ± SD] of the nematode species *Plectus similis* and *Plectus* sp. fed with different diets, i.e., *L. pneumophila* (Corby wild-type, ProA-lacking Corby Δ*proA* mutant and its respective complementary mutant Corby Δ*proA proA*), *E. coli* BL21 (BL21, ProA-producing strain) and *E. coli* OP50. Additionally, nematodes were incubated for 2 h in ProA prior to feeding on *E. coli* OP50 (OP50 + ProA). Statistical differences according to the Dunn test (three *L. pneumophila* strains) and Mann–Whitney U test (two *E. coli* OP50 and two *E. coli* BL21 strains), respectively. * *p* < 0.05.

**Figure 4 microorganisms-11-00738-f004:**
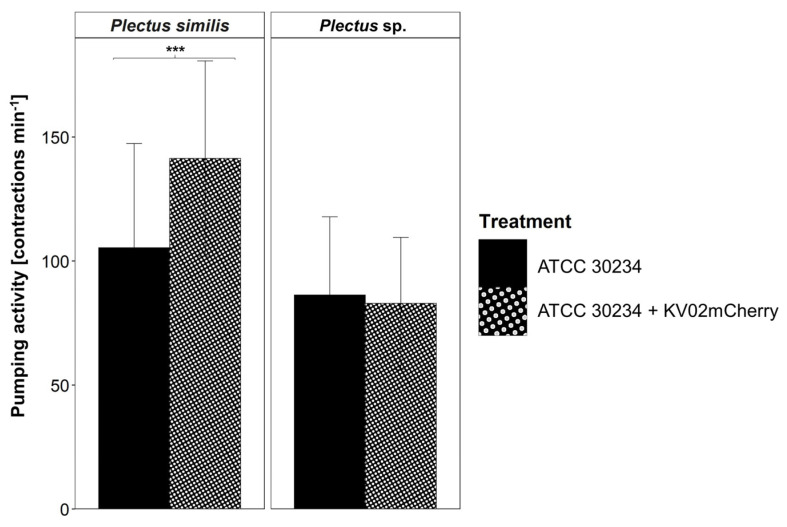
Pharyngeal pumping activity per individual [contractions min^−1^ ± SD] of the nematode species *Plectus similis* and *Plectus* sp. fed with the *A. castellanii* strain ATCC 30234. *A. castellanii* were infected with *L. pneumophila* KV02mCherry (cooling tower isolate) and fed to nematodes 24 h after infection. Statistical differences according to the Mann–Whitney U test. *** *p* < 0.001.

**Figure 5 microorganisms-11-00738-f005:**
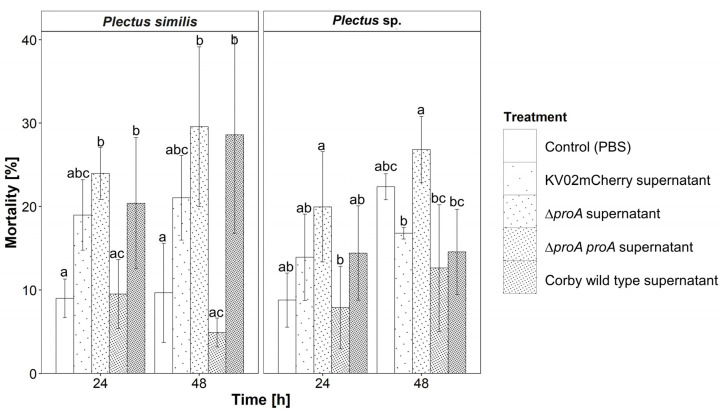
Mortality [% ± SD] of the nematode species *Plectus similis* and *Plectus* sp. after incubation with different supernatants of *Legionella* for 24 and 48 h. The strains tested were *L. pneumophila* KV02mCherry (cooling tower isolate), *L. pneumophila* Corby wild-type and a *L. pneumophila* Corby Δ*proA* mutant and its respective complementary mutant strain *L. pneumophila* Corby Δ*proA proA*. Control = PBS buffer. Statistical differences according to Tukey’s post hoc test (*p* < 0.05). Bars with the same letter are not statistically different according to Dunn’s post hoc test (*p* < 0.05).

**Table 1 microorganisms-11-00738-t001:** Co-occurrence of nematode genera (Ind. 100^−1^ mL ± SD) and *Legionella pneumophila* (+, detection; -, no detection) in natural and technical water habitats. Nematode genera with assigned *c-p* values [23] arranged by trophic group.

Trophic Group	Genus	*c-p*	Swimming Lakes (*n* = 9)			Cooling Towers (*n* = 7)						
			Water Surface, Algae	Read, Macrophytes	Submerged Stones, Litter	CT 1	CT 2	CT 3	CT 4	CT 5	CT 6	CT 7
Plant feeders	*Aphelenchoides*	2	1.5 ± 1.3	2.8 ± 4.0	1.7 ± 3.0	-	-	-	-	-	-	-
	*Filenchus*	2	-	-	3.5 ± 6.0	21.5	-	-	25.6	-	-	-
Bacterial feeders	*Acrobeloides*	2	-	-	-	-	-	28.6	76.9	-	-	-
	*Alaimus*	4	1.5 ± 1.3	2.1 ± 3.6	-	-	-	-	-	-	-	-
	*Bastiania*	3	0.7 ± 1.2	-	-	-	-	-	-	-	-	-
	*Chromadorina*	3	63.3 ± 36.7	136.2 ± 64.2	68.2 ± 45.0	-	-	-	-	-	-	-
	*Diplogasteritus*	1	-	-	-	-	-	-	-	9.5 ± 16.5	-	-
	*Diploscapter*	1	-	-	-	-	-	-	-	161.9 ± 16.5	-	-
	*Eucephalobus*	2	-	-	-	-	-	-	-	-	-	28.6
	*Eumonhystera*	1	16.1 ± 17.3	6.6 ± 5.7	1.1 ± 0.9	-	-	-	-	-	-	-
	*Heterocephalobus*	2	0.4 ± 0.7	-	-	10.8	-	-	-	-	-	-
	*Monhystrella*	1	0.7 ± 1.2	-	-	-	-	-	-	-	-	-
	*Panagrolaimus*	1	2.2 ± 2.1	-	-	-	-	-	-	-	-	-
	*Paraphanolaimus*	3	-	-	2.4 ± 4.1	-	-	-	-	-	-	-
	*Plectus*	2	3.4 ± 4.2	1.5 ± 1.3	1.1 ± 0.9	1044.6	-	28.6	282.1	-	28.6	-
	*Rhabdolaimus*	3	-	18.8 ± 32.5	7.0 ± 9.2	-	-	-	-	-	-	-
Omnivores	*Achromadora*	3	0.8 ± 1.3	-	6.1 ± 6.1	-	-	-	-	-	-	-
	*Epidorylaimus*	4	-	-	1.2 ± 2.1	-	-	-	-	-	-	-
	*Eudorylaimus*	4	0.7 ± 1.2	-	-	-	-	-	-	-	-	-
	*Laimydorus*	4	7.7 ± 13.3	38.3 ± 64.2	1.0 ± 1.7	-	-	-	-	-	-	-
	*Mesodorylaimus*	4	6.0 ± 9.4	88.3 ± 150.8	4.2 ± 3.8	-	-	-	-	-	-	-
Predators	*Ironus*	4	-	2.2 ± 3.6	-	-	-	-	-	-	-	-
	*Mononchus*	4	-	0.8 ± 1.4	-	-	-	-	-	-	-	-
	*Tobrilus*	3	60.7 ± 49.2	12.4 ± 10.7	17.9 ± 25.0	-	-	-	-	-	-	-
Total number of nematode genera			14	11	12	3	0	2	3	2	1	1
Detection *Legionella pneumophila*			-	-	-	+	+	+	+	+	- ^1^	+

^1^ Biocide shock dosage before sampling.

**Table 2 microorganisms-11-00738-t002:** Nematode activity [% ± SD] of *Plectus similis* and *Plectus* sp. fed with different diets, i.e., *L. pneumophila* (cooling-tower isolate KV02mCherry, Corby wild-type, ProA-lacking Corby Δ*proA* mutant and its respective complementary mutant Corby Δ*proA proA*), *E. coli* OP50 (OP50, OP50 mixed with KV02mCherry filtrate), *E. coli* BL21 (BL21, ProA-producing strain) and non-infected and *Legionella* (KV02mCherry)-infected *A. castellanii*. Additionally, nematodes were incubated for 2 h in ProA prior to feeding on OP50 (OP50 + ProA). Statistical differences according to Dunn’s post hoc test (*p* < 0.05) were examined between activity modes (i.e., columns) for each diet separately. Data with the same or no letters are not statistically different.

Strain	*Plectus similis*			*Plectus* sp.		
	Inactive	Active	Feeding	Inactive	Active	Feeding
*L. pneumophila*						
KV02mCherry	46 ± 17 ab	54 ± 17 ab	9 ± 5	77 ± 9	23 ± 9	14 ± 2
Corby wild-type	59 ± 12 a	41 ± 12 a	7 ± 3	71 ± 24	29 ± 24	13 ± 1
Δ*proA*	33 ± 8 ab	67 ± 8 ab	7 ± 1	68 ± 16	32 ± 16	14 ± 2
Δ*proA proA*	17 ± 4 b	83 ± 4 b	16 ± 5	59 ± 3	41 ± 3	12 ± 0
*E. coli* OP50						
OP50	12 ± 1 a	88 ± 1 a	87 ± 1 a	52 ± 9	48 ± 9	39 ± 2 a
OP50 + KV02mCherry filtrate	21 ± 9 ab	79 ± 9 ab	21 ± 2 ab	53 ± 8	47 ± 8	11 ± 1 b
OP50 + ProA	76 ± 3 b	24 ± 3 a	12 ± 3 b	66 ± 8	34 ± 8	21 ± 2 ab
*E. coli* BL21						
BL21 pET22b(+)-*proA*	46 ± 6	54 ± 6	51 ± 6	41 ± 4	59 ± 4	54 ± 6
BL21	54 ± 2	46 ± 2	44 ± 3	60 ± 11	40 ± 11	38 ± 12
*A. castellanii*						
ATCC 30234	56 ± 7	44 ± 7	41 ± 6	38 ±12	62 ± 12	52 ± 18
ATCC 30234 + KV02mCherry	46 ± 10	54 ± 10	49 ± 11	31 ±4	69 ± 4	58 ± 8

## Data Availability

Data are contained within the article.

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
