# Peer review of "Legionella pneumophila* and Free-Living Nematodes: Environmental Co-Occurrence and Trophic Link"

_microorganisms, 2023, doi:10.3390/microorganisms11030738_

Round 1

Reviewer 1 Report

The manuscript is interesting but some concerns need to considered:

1- the abstract need to rewrite to include the merit of the work.

2- the aim of the introduction reconstruction to include the work done

3- the discussion is long and many sentences are repeated, please reconsider it.

Reviewer 2 Report

This paper describes the relationship between L. pneumophila and two nematodes isolated from the environment and the role of its secretory protein ProA. Three hypotheses were investigated concerning the co-occurrence of L. pneumophila and nematodes in biofilms originating from lakes and cooling tower, the source of feeding and the role of metabolites on nematode fitness. The authors demonstrated the presence of both organisms in biofilms and the diet preference of nematodes, that do not appreciate L. pneumophila. The role of ProA varied with diet and nematode species. Finally, the Trojan horse-like transmission of L. pneumophila from Amoeba to nematodes was demonstrated.

The overall quality of the manuscript is good and data support the conclusions. I have some minor comments and two major comments about figures 3 and 5.

Minor comments:

L. 66-67 : “A major role for the pathogenesis and ecology of L. pneumophila play secreted secondary metabolites.” I do not understand this sentence. I suggest to replace the verb by “comes from”: A major role for the pathogenesis and ecology of L. pneumophila comes from secreted secondary metabolites.

L. 69-70: “T4SS secretes more than 300 effector proteins, which manipulate pivotal host processes including autophagy, death pathways, protein translation and turnover and innate immunity [3,26].” Wrong citation: ref 26 refers to T2SS, not to T4SS. Ref 26 cites ref. 3, so ref 26 should be suppressed for this assertion, but can be added at the end of the next sentence with ref. 27-28.

L. 130: “In each of these lakes, the biofilms on in three sub-habitats were sampled “ : “on” should be suppressed.

L. 149-150: “An alignment of 99.86% confirmed the isolate KV02 to belong to the species L. pneumophila.” This sentence is a result and should be placed in the corresponding section.

L. 196-198: “L. pneumophila KV02mCherry were adjusted to 3 x 104 colony forming units /ml (cfu/ml) in amoeba buffer”: ameba buffer is not described unless the authors mean PAS buffer? In that case, I suggest to write “in PAS buffer”. Why PBS was not used to adjust bacterial concentration as described l. 207-208?

L. 218: “onto a assay plate”: replace “a” by “an”.

L. 225-241: These sentences are results, not methods and should be placed in the result section.

L. 283-287: this part should be suppressed as the description of the supernatant preparation was already described l.258-263.

L. 291&292&294: “tab water” : do the authors mean tap water?

L. 342&377&473: “(Figure 1): ”: “:” should be replaced by “.”

L. 530: wrong reference as ref 61 is about amoebae, not nematodes. Please change the ref.

L. 566: Please replace (13) by [13].

Major comments:

Figure 3 interpretation: Due to the biological variability, significant differences could be obtained only with non-parametric tests based on the median. I recommend to indicate a tendency to decrease rather than a significant difference (to be applied in the result and discussion sections). When I compared the figure to the text, I do not fully agree with conclusions. L.402-403: “Indeed, the observed pumping frequencies of P.  similis (75 contractions min-1) and Plectus sp. (25 contractions min-1) were comparable to the two other L. pneumophila strains.” I would rather say: Pumping frequency of P. similis in presence of the complement mutant strain L. pneumophila Corby ΔproA proA (75 contractions min-1) was in-between the frequency in presence of the wild strain (61 contractions min-1) and the frequency with the Corby ΔproA mutant (100 contractions min-1). This tendency was not observed for Plectus sp., as the frequency in presence of both mutant strains was identically reduced to 25 contractions min-1.

L. 445-446: Why was the adaptation time different for Legionella (only 3 h)? Please justify.

Figure 5: L. 494&499, strains should be replaced by supernatants to avoid confusion. I recommend to avoid to drive conclusion for Plectus sp. at 48h as the mortality in PBS was very high and to say that data at 48h were not considered due to the high mortality in the control (L. 508-509). So as well remove “and 48h…” L. 648-649 in the discussion.

l. 636: Looking at figure 3, please add: “but not with E. coli BL21 pET22b(+)-proA.” after “In contrast, Plectus sp. decreased the pumping rate when fed with the L. pneumophila strains ΔproA and ΔproA proA”.

L. 688: To complete the discussion on Trojan horse strategy, I suggest to add the hypothesis that L. pneumophila could hide from nematode detection inside the amoeba and by the way could access to a new host without being detected before ingestion.

Round 2

Reviewer 1 Report

The author responds to the comments.